# Spatial–Temporal Patterns of Sympatric Asiatic Black Bears (*Ursus thibetanus*) and Brown Bears (*Ursus arctos*) in Northeastern China

**DOI:** 10.3390/ani12101262

**Published:** 2022-05-14

**Authors:** Yunrui Ji, Fang Liu, Diqiang Li, Zhiyu Chen, Peng Chen

**Affiliations:** 1Ecology and Nature Conservation Institute, Chinese Academy of Forestry, Beijing 100091, China; jiyunrui-caf@foxmail.com (Y.J.); lidq@caf.ac.cn (D.L.); 2Key Laboratory of Biodiversity Conservation, State Forestry and Grassland Administration, Beijing 100091, China; 3The Administration of Duluhe Provincial Wetland Nature Reserve, Hegang 154100, China; vangogh22222@163.com; 4College of Landscape Architecture, Northeast Forestry University, Harbin 150040, China

**Keywords:** interspecific relationship, niche differentiation, coexistence, camera traps, temporal activity pattern

## Abstract

**Simple Summary:**

The mechanisms of coexistence between large carnivores are critical for conservation and the management of endangered species. In northeastern China, the distributions of Asiatic black bears (*Ursus thibetanus*) and brown bears (*Ursus arctos*) overlap widely. To better understand the mechanisms of coexistence among sympatric Asiatic black bears and brown bears, we assessed their spatial–temporal patterns using camera trapping data in Taipinggou National Nature Reserve, Heilongjiang Province, China. We found evidence for spatial and temporal divergences among the two Ursidae species. Asiatic black bears avoided brown bears by occupying higher elevations and being more diurnal where they coexisted. This study demonstrated how two bear species coexist through spatial and temporal niche separation, which will facilitate future studies on the mechanisms determining the coexistence of species and provide new basic data for animal conservation.

**Abstract:**

Studying the spatial and temporal interactions between sympatric animal species is essential for understanding the mechanisms of interspecific coexistence. Both Asiatic black bears (*Ursus thibetanus*) and brown bears (*Ursus arctos*) inhabit northeastern China, but their spatial–temporal patterns and the mechanism of coexistence were unclear until now. Camera traps were set in Heilongjiang Taipinggou National Nature Reserve (TPGNR) from January 2017 to December 2017 to collect photos of the two sympatric bear species. The Pianka index, kernel density estimation, and the coefficient of overlap were used to analyze the spatial and temporal patterns of the two sympatric species. Our findings indicated that the spatial overlap between Asiatic black bears and brown bears was low, as Asiatic black bears occupied higher elevations than brown bears. The two species’ temporal activity patterns were similar at sites where only one species existed, yet they were different at the co–occurrence sites. Asiatic black bears and brown bears are competitors in this area, but they can coexist by changing their daily activity patterns. Compared to brown bears, Asiatic black bears behaved more diurnally. Our study revealed distinct spatial and temporal differentiation within the two species in TPGNR, which can reduce interspecific competition and facilitate coexistence between them.

## 1. Introduction

Studies on the coexistence of sympatric species are important in the context of ecological research, which is conducive to understanding the maintenance mechanism of community diversity [1,2,3]. In 1917, Grinnell first used the term “niche” to describe the distribution unit of species [1]. Space, time, and food are three typical components of ecological niches [4,5]. The interspecific overlap in the above three dimensions may imply interspecific competition [6,7]. To reduce interference and competition, coexisting species would divide resources across different niches [8,9,10,11].

Space and time are the most common and important dimensions for niche separation [6,12]. In terms of space, dominant animals may squeeze out secondary competitors through direct methods (occupying resource–rich habitats) and indirect ones (leaving traces, such as footprints, feces, urine, etc.) [10,13,14,15]. Compared with the spatial niche, temporal partitioning is more flexible and elastic [16]. Sympatric species can change their activities and behavior patterns on different temporal scales (daily, monthly, and seasonally) to mitigate interference and competition [17,18,19]. Animal regional coexistence is a relatively stable state that has evolved over time [12]. Spatial use and activity patterns of sympatric species can be influenced by several factors, including prey availability, human disturbance, and climate [20,21,22,23,24]. The altered spatial–temporal interactions between sympatric species in communities could have rippling effects throughout the entire ecosystem [25,26]. Assessing the mechanisms of interspecific coexistence is important for conservation and the management of endangered species [27].

Among the eight species of bears in the world, four are known to live in China, including the giant pandas (*Ailuropoda melanoleuca*), brown bears (*Ursus arctos*), Asiatic black bears (*Ursus thibetanus*), and sun bears (*Helarctos malayanous*) [28]. Both giant pandas and sun bears have narrow distributions in China, yet the former is endemic to China and differed with the latter [29,30,31,32]. Additionally, Asiatic black bears and brown bears are widespread, and their distributions overlap in southwestern and northeastern China [29,30]. The two species coexist principally through habitat divergence in southwestern China: brown bears often use alpine meadows while Asiatic black bears prefer forests [29,30,33]. However, both of them are large forest–dwelling omnivorous animals with a plant–based diet in northwestern China [29,30]. Although studies related to the distributions [33,34], diets [35,36,37], habitats [8,35,38], and activities [35,39,40,41] of the Asiatic black bears and brown bears have been conducted individually, research on interactions between them is deficient. To fill this knowledge gap, we quantified the spatial and temporal activity patterns and habitat use of Asiatic black bears and brown bears based on infrared camera–trap surveys conducted in the Taipinggou National Nature Reserve (TPGNR), aiming to investigate their spatial–temporal patterns and explore the mechanisms of coexistence for the two sympatric carnivores. We hypothesized that Asiatic black bears, as the subordinate competitors, would avoid brown bears in the spatial and temporal niches to minimize the probabilities of encounters with the latter. We also aimed to explore the main driving factors that assisted the coexistence between the two sympatric bear species in order to better manage and conserve them.

## 2. Materials and Methods

### 2.1. Study Area

The TPGNR is located in Heilongjiang Province, northeastern China (48°02′48″~48°20′19″ N, 130°31′12″~130°50′11″ E), at elevations ranging from 72 m to 556 m above sea level (Figure 1). It occupies an area of 22,199 hm^2^ and is bordered to the east by Russia across the Heilongjiang River [42]. The area is in the middle temperate zone, with a temperate continental monsoon climate [43]. The TPGNR is mainly covered by natural secondary forest, which is regarded as representative of the temperate forest ecosystems along the China–Russian border [42]. The TPGNR provides important habitats for both Asiatic black bears and brown bears and plays a vital role in the cooperation of wildlife conservation across the China–Russian border [42].

### 2.2. Camera Trapping Surveys

We divided the study area into 2 × 2 km grids. Considering topographic features, vegetation types, and the signs of bear activity, a total of 20 camera traps (one camera trap inside each grid cell) were set up in the TPGNR to monitor bears in January 2017 (Figure 1). The distance between any two camera traps was greater than 2 km. Although the effect of using bait on animal movement and the habitat is contested and varies with species [44,45,46], attractants, such as food or glandular scents, are often applied to increase the likelihood of capturing animals when the target species are too fast or small to be captured by cameras [44,45]. Hence, we set up each camera trap composed of three infrared cameras focused on a central bait of meat from three different aspects [44,45,46,47].

The angle between each pair of infrared cameras was approximately 120°. The bait was 5~7 kg of lamb, which was hung above the central location at 3 m high. The cameras were fastened to trees at a height of 80~100 cm above the ground and were set to be active constantly per day. The location, altitude, slope, and aspect of each trap were recorded. Images taken by camera traps were collected every 2 months for 1 year until December 2017.

Consecutive photographs of the same species at the same site were deemed independent when there was at least a 30-min interval between them [48]. We calculated the relative abundance index (RAI), which was defined as the number of independent detections per 100 days of camera trapping for each species [49]. We defined co–detections when both species were detected at the same camera location; otherwise, a location was labeled as empty or single–species detection.

### 2.3. Spatial Distribution Overlap and Habitat Use

To investigate the spatial overlap between Asiatic black bears and brown bears, we calculated the relative abundance index (RAI) [48] of each species in each camera–trapping site. We considered each camera trapping site as spatially independent and used the RAI to calculate Pianka’s O index [50]. Pairwise species spatial overlaps of the two species were analyzed by R [51] using the “spaa” package [52]. The closer to 1 the Pianka’s O index was, the higher degree of overlap of species in the spatial distribution.

To assess the habitat use of the two species, we divided the altitude and vegetation types into 2 levels and divided the aspect and slope into 3 levels (Table 1). Habitat use by the Asiatic black bear and brown bear was evaluated using Bailey’s 95% simultaneous confidence intervals [53].

### 2.4. Temporal Overlap

We explored the temporal overlap between Asiatic black bears and brown bears at three scales (monthly, seasonal, and daily). The number of independent captures was summarized by month and season, and the monthly relative abundance index (MRAI) and seasonal relative abundance index (SRAI) were calculated in accordance with the previous study [17], whose variation can perform the monthly and seasonal activity pattern of the species [17]. According to the local climate, seasons were divided by the following criteria: spring (April–May), summer (June–August), autumn (September–October), and winter (November–March of the following year) [43]. The Wilcoxon signed–rank test was used to analyze the differences between the monthly and seasonal activity patterns of the Asiatic black bears and brown bears.

The daily activity rhythms of the species were measured based on the non–parametric circular kernel density models, considering independent detection as random sampling from continuous activity [54]. The activity rhythm can be separated into activity level and activity pattern in the kernel density models [55,56]. The activity level is the ratio of the areas under and above the curve of the circular probability density function f (x), representing a percentage of time active, while activity pattern is the shape and trend of the curve [55,56]. The package “activity” in R (R Core Team, Vienna, Austria) was used to fit circular kernel density models for Asiatic black bears and brown bears to estimate their activity levels by 10,000–times smoothed bootstrapping [57]. Subsequently, the randomization test and Wald’s test were carried out to detect the differences in activity patterns and activity levels between the two species [55,57]. We compared the daily activity rhythm and overlap using co–detection and single–species detection. The coefficient of overlap (Δ) ranged from 0 (no overlap) to 1 (complete overlap) and was obtained by taking the minimum of the density functions of the two cycles being compared at each time point [55], which was calculated using the “overlap” package [58] in R.

## 3. Results

### 3.1. Spatial Distribution Overlap and Habitat Use

A total of 5034 trapping nights were conducted at the 20 camera–trapping sites. We obtained 127 independent detections of Asiatic black bears across 15 sites and 105 of brown bears across 15 sites, respectively (Figure 1). The two species co–occurred at 10 camera–trapping sites (Figure 1), and the spatial overlap between them was low based on Pianka’s O index of 0.268 (CI: 0.116–0.474, SD = 0.095). Both species preferred sunny slopes, steep slopes (≥15°), and deciduous broad–leaved forests (Table 2). Brown bears showed statistically different use of the elevation compared with Asiatic black bears, preferring to occupy areas below 300 m (Table 2).

### 3.2. Temporal Overlap

#### 3.2.1. Monthly and Seasonal Activity Patterns

The monthly activity patterns of the two bear species were not significantly different (v =24, *p* = 0.759; Figure 2A), and they both showed bimodal activity patterns throughout the year. Asiatic black bears’ MRAI peaked in May and August, respectively, while brown bears’ MRAI stood the highest in May and September (Figure 2A). They were rarely detected from November until April, yet only one brown bear was detected in February. Asiatic black bears behaved similarly to brown bears in seasonal activity patterns (v = 7, *p* = 0.625; Figure 2B), with the highest SRAI in spring and the lowest SRAI in winter (Figure 2B).

#### 3.2.2. Daily Activity Pattern

Circular kernel density models based on single–species detection indicated similar daily activity patterns (*p* = 0.93) and levels (W = 0.121 ± 0.019; *p* = 0.889) between Asiatic black bears and brown bears (Figure 3A). Both species had three activity peaks, two of which nearly overlapped (5:00–6:00 and 19:00–20:00; Figure 3A). Conversely, they had significantly different daily activity patterns (*p* = 0.01) at the co–occurred sites, with separate activity peaks and Asiatic black bears being more diurnal than brown bears (Figure 3B). Moreover, a much lower overlap was observed between them at the co–occurred sites (Δ = 0.737; CI: 0.589–0.884; Figure 3B) than where a single species occurred (Δ = 0.826; CI: 0.702–0.911; Figure 3A). Nevertheless, their daily activity levels were similar (W = 1.221 ± 0.137; *p* = 0.269; Figure 3B).

## 4. Discussion

In forest ecosystems, elevation is one of the main driving factors for spatial heterogeneity, which is of great importance to the regional coexistence of carnivores [12,26]. We found that brown bears had a low spatial overlap with Asiatic black bears through occupying lower elevations, similar spatial separation mechanisms also occurred within other carnivores, such as felids [59,60]. Nonetheless, their preferences for elevation in the TPGNR are diverse to the previous studies in America, where grizzly bears range over a broader spectrum of elevations and use the lower and higher elevations more frequently than American black bears (*Ursus americanus*) [61]. This inconsistency may be attributed to the spatial discrepancy in food availability, which can influence the bears’ distribution and habitat use [62,63]. More research should be undertaken to investigate the relationship between food availability and the habitat use of bears in northeast China.

Our findings indicated that the two bear species had a similar habitat preference on the slope, aspect, and vegetation types, which are contrary to previous studies where differences in habitat use between American black bears and grizzly bears were evident [61,64]. A possible explanation for this may be the seasonal movements of bears, which are necessary for their survival in mountain habitats. In this study, the consistency of habitat preference between them was taken from the overall one–year performance, yet it can vary across seasons, as studies have shown in Japan [64,65]. Whereas, we could not test for potential differences in habitat use among seasons due to the short survey time, which needs to be investigated by future research.

Numerous animals can modify their activity patterns with seasonal variation, which is an evolutionary adaptation to nature [23,56,66]. The ability to hibernate is arguably the best example of the phenotypic plasticity shown by mammals [23]. Bears became inactive from the end of December to the beginning of April in the TPGNR, consistent with findings in other temperate regions [35,39,67]. Whereas, a brown bear was photographed by an infrared camera at the end of February. A similar phenomenon was also recorded in Tangjiahe National Nature Reserve, Sichuan, where fresh droppings of an Asiatic black bear were found in March [35]. Bears can wake up and respond quickly during hibernation if needed (e.g., reacting to abnormal climate and damage in their den), which is their notable adaptation to the environment [68]. Furthermore, brown bears could emerge from dens to forage on vegetation slightly earlier than black bears to minimize interspecific competition [69].

Asiatic black bears and brown bears have similar seasonal activity patterns, with activity levels being lower before and after hibernation than in other periods, as found previously [41,66,70]. Nevertheless, we found that they were more active in spring than in autumn, which differed from previous studies where they were more active in autumn [41,71]. These differences may be attributed to the less succulent vegetation in the TPGNR during spring than in Taiwan and coastal British Columbia, which are mainly dominated by rainforests [41,71]. Asiatic black bears and brown bears have to forage longer to meet their nutritional demands in the TPGNR than in warmer areas. Both species had an active peak in the oak season, yet their activity peaks were staggered by month, with the latter peaking a little later. Furthermore, brown bears were much less active in September than in May, which differed from a previous study where they were more active in July and September due to the arrival and death times of pink salmon (*Oncorhynchus gorbuscha*) and chum salmon (*Oncorhynchus keta*) in Bish Creek, respectively [71]. This difference may be attributed to the use of bait. Brown bears are less likely to occupy areas with bait in the TPGNR when natural food is abundant. Although numerous studies have found that bait does not affect the temporal pattern, habitat use, and movement of the animals [72,73,74], it is unclear whether the bait influences the brown bears’ behavior in the TPGNR, which needs further work to explore.

We found that the two species’ temporal patterns were similar in the overall daily, monthly, and seasonal activity. However, temporal divergence happened between them when they co–occupied habitats, with Asiatic black bears tending to be more diurnal. These variations in their activity patterns can facilitate their coexistence. Similar segregation mechanisms have been observed in North America [70,75], where American black bears are subordinate to brown bears and consequently adjust their temporal activity to avoid conflicts with grizzly bears [70]. In the carnivore community, dominant carnivores are mainly nocturnal and twilight, while secondary competitors perform diurnal activities to avoid disturbing suppression from the dominant species [12]. For example, leopards (*Panthera pardus*) are more active diurnally than tigers (*Panthera tigris*) in the Changbai Mountains, China [76]. In addition, human activities have had significant impacts on the rhythm of animal activities and their interspecific relationships [11,21,77]. Both American black bears and grizzly bears could adjust their activity patterns (e.g., being more nocturnal in areas where human activity is high) to achieve coexistence with humans [40,70,75]. The continuous expansion of human activity will bring new challenges to species interactions globally [11,21]. Further work needs to be carried out on how Asiatic black bears and brown bears coexist under the influence of human disturbance in China.

## 5. Conclusions

Asiatic black bears and brown bears coexist in northeastern China, where they compete for resources due to their biological and ecological similarities [30]. Our research provides insights into spatial and temporal associations between the species and tentative evidence that both of them could change their behaviors to achieve coexistence. In terms of spatial niche, brown bears inhabit areas with lower elevations than Asiatic black bears. Temporal niche separation between them was also obvious, with Asiatic black bears behaving more diurnally when they co–occupied habitats.

Large carnivores are the flagship species for the conservation of habitats or ecosystems [78]. Results from this study have key implications for bear management, the maintenance of complex communities, and biodiversity conservation. Our map of bear distribution will facilitate the development of spatially explicit conservation recommendations and prioritization of areas for the bears in the TPGNR. Understanding the spatial–temporal pattern of two bear species can assist local residents and wildlife managers in reducing the risk of bear encounters. The communities around the TPGNR have a small and aging population, but villagers collect mushrooms and firewood in the forest, which can increase the risk of human–bear encounters. The bears’ activity peaks in May and August–September is when humans should reduce their activities in the wild. The wildlife manager should strengthen the management of human activities at this time. Moreover, deciduous broad–leaved forests are the commonly preferred habitats of Asiatic black bears and brown bears; thus the staff at the TPGNR should be particularly cautious when patrolling these forests. During the day, the field staff should avoid working in the woods in the early morning and at dusk.

Previous studies have suggested that human activity could affect the temporal patterns of animals [6,70,79]. A better understanding of the activity patterns of carnivores can guide the implementation of protection management, hence reducing the human disturbance to bears. Additionally, the interspecific relationship of carnivores is not fixed, which can be affected by changes in the environment, human disturbance, and carnivorous population structure [21,76,77]. Furthermore, research is needed to explore the issue of how large carnivores adapt to this change. The inclusion of spatial–temporal behaviors in conservation and management plans can help to promote the coexistence of sympatric species.

## Figures and Tables

**Figure 1 animals-12-01262-f001:**
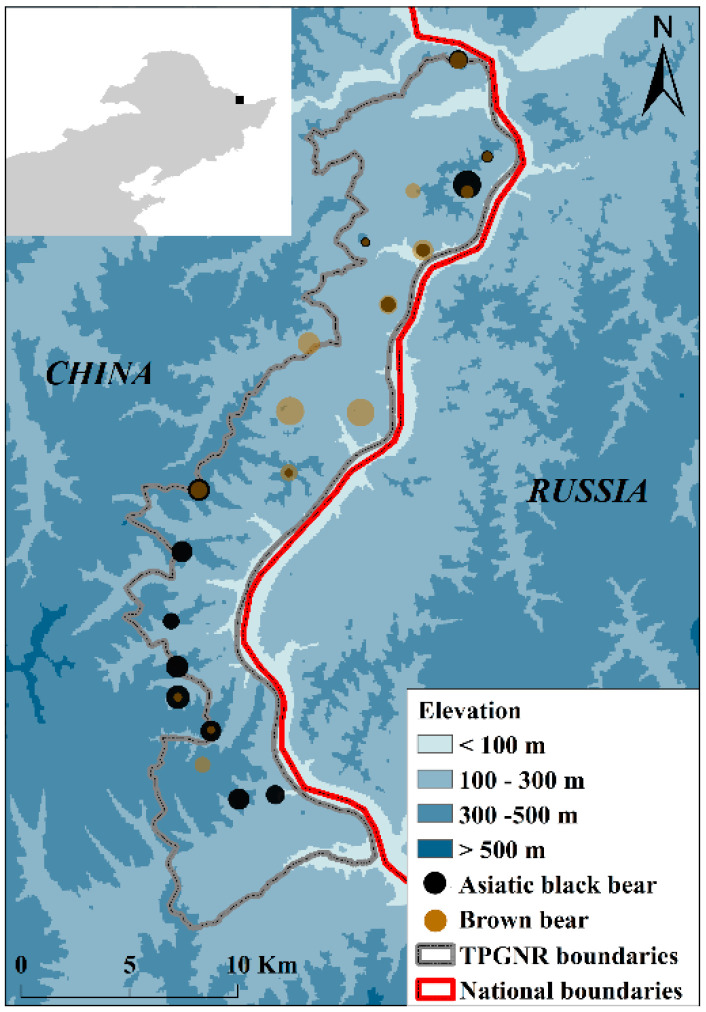
Study area and camera–trapping sites indicating where Asiatic black bears and brown bears exist in Taipinggou National Nature Reserve, Heilongjiang Province, China, 2017. The black and brown points show the center of each camera traps, and their size indicated the relative abundance index of Asiatic black bears and brown bears in independent camera traps.

**Figure 2 animals-12-01262-f002:**
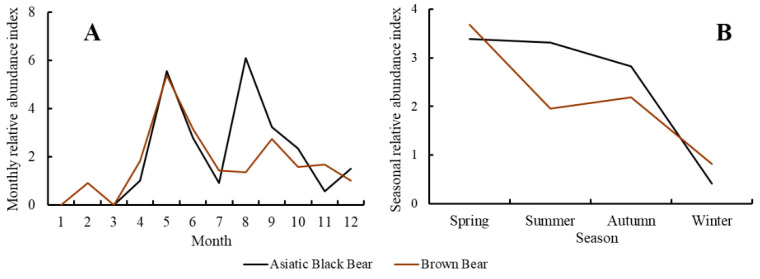
Monthly (**A**) and seasonal (**B**) activity patterns of Asiatic black bear and brown bear in Taipinggou National Nature Reserve, Heilongjiang Province, China, 2017.

**Figure 3 animals-12-01262-f003:**
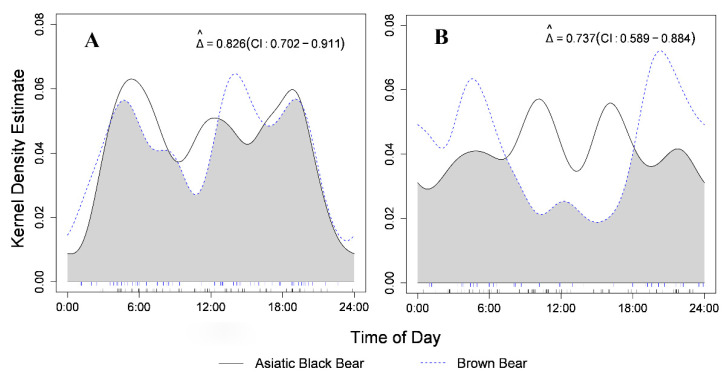
Daily activity patterns of Asiatic black bear and brown bear based on single–species detection (**A**) and co–detection (**B**) in Taipinggou National Nature Reserve, Heilongjiang Province, China, 2017. The mean value of the coefficient of overlap $\hat{\Delta }$ is represented by the light grey area under the curves. Single–species detection was defined as independent detection of Asiatic black bear and brown bear at camera–trapping sites where only one species had occurred.

**Table 1 animals-12-01262-t001:** Classification criteria of habitat variables for Asiatic black bear and brown bear in Taipinggou National Nature Reserve, Heilongjiang Province, China, 2017.

Habitat Variables	Classification Criteria
Altitude (H_alt_, m)	Level 1: 170 m < H_alt_ ≤ 300 m
	Level 2: 300 m < H_alt_ ≤ 441 m
Aspect	Type 1: Half–sunny slope (45°~135°, 225°~315°)
	Type 2: Shady slope (0°~45°, 315°~360°)
	Type 3: Sunny slope (135°~225°)
Slope	Level 1: <5°
	Level 2: 5°~15°
	Level 3: ≥15°
Vegetation type	Type 1: Coniferous and broadleaved mixed forest

**Table 2 animals-12-01262-t002:** Habitat use of Asiatic black bear and brown bear in Taipinggou National Nature Reserve, Heilongjiang Province, China.

Habitat Factor	Classification Criteria	Expected Proportion Used (P_io_)	Actual Proportion Used (P_i_)		Bailey’s 95% Interval for P_i_		Preference	
			Brown Bear	Asiatic Black Bear	Brown Bear	Asiatic Black Bear	Brown Bear	Asiatic Black Bear
Altitude	Level 1	0.55	0.667	0.467	0.625–0.692	0.426–0.491	+	–
	Level 2	0.45	0.333	0.533	0.295–0.360	0.491–0.558	–	+
Aspect	Type 1	0.25	0.200	0.200	0.168–0.231	0.168–0.231	–	–
	Type 2	0.50	0.333	0.467	0.295–0.360	0.426–0.491	–	–
	Type 3	0.25	0.467	0.333	0.426–0.491	0.295–0.360	+	+
Slope	Level 1	0.25	0.200	0.267	0.168–0.231	0.231–0.295	–	0
	Level 2	0.60	0.600	0.533	0.558–0.625	0.491–0.558	0	–
	Level 3	0.15	0.200	0.200	0.168–0.231	0.168–0.231	+	+
Vegetation type	Type 1	0.25	0.133	0.200	0.105–0.168	0.168–0.231	–	–

Avoidance “–” is indicated when the available proportion of habitat type is higher than upper value of the confidence limit; preference “+” is indicated when the available proportion of habitat type is lower than lower value of the confidence limit. Random “0” is indicated when the available proportion of habitat type is included by 95% confidence intervals.

## Data Availability

The data presented in this study are available on request from the corresponding author. The data are not publicly available due to privacy restrictions.

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
