# Peer review of "Spatial–Temporal Patterns of Sympatric Asiatic Black Bears (Ursus thibetanus) and Brown Bears (Ursus arctos) in Northeastern China"

_animals, 2022, doi:10.3390/ani12101262_

Round 1

Reviewer 1 Report

This is an interesting study on the spatial and temporal interactions between sympatric Asiatic black bears and brown bears which both inhabit northwestern China.

The authors, by means of camera traps found that the spatial overlap between Asiatic black bears and brown bears was low, and that the two species' temporal activity patterns were similar at sites where only one species existed and that they were partitioned at the co-occurrence location. The two bear species can therefore coexist by changing their daily activity patterns. Compared to brown bears, Asiatic black bears behaved more diurnally.

The study revealed apparent spatial and temporal partitioning within the two species. The results are interesting and very useful for conservation actions.

The references are appropriate, and the table and the graphs are also appropriate.

Maybe the the formulas for Pianka’s O index…is not necessary and the authors could simply cite the reference of the formula….

For the rest I do not have more comments as the statistic is appropriate and the discussion of the results are well supported by the results.

I think that this is a paper which can be quite useful for the conservation of the two species and is also paving the base for other investigations on other species which coexis in sympatry.

Author Response

Response to Reviewer 1 Comments

Thank you for your positive comments and valuable suggestions to improve the quality of our manuscript. According to your suggestions, we have revised our manuscript. The detailed point-by-point responses are listed below.

Points 1 Maybe the formulas for Pianka’s O index…is not necessary and the authors could simply cite the reference of the formula….

Response: Thank you for this suggestion, and we have deleted the formulas for Pianka’s O index.

Reviewer 2 Report

Review of the paper „Spatial-temporal pattern of sympatric Asiatic black bears (Ursus thibetanus) and brown bears (Ursus arctos) in northeastern China”

The article deals with the interesting topic of the coexistence of large predators in space. Although it is very important and I have read the text with interest, I believe that the way of writing requires some effort before it is published. I encourage authors to think about what they want to write and when to use the word similarity and when to differentiate. Below, I propose a few changes that could help you stay consistent in the line of thinking.

Simple summary and abstract should be re-written and below example comments concern also the whole paper. The biggest problem is inconsistency in the arguments explaining when territories overlap or no, as well as contradictory information that often appears directly in adjacent sentences.

coexistence or co-existence?

L19 – I do not understand what it means that the territories overlap significantly, specifically the word significantly, which is reserved for the results of statistical tests. At this point, readers should find out whether the overlap is only spatial or qualitative, e.g. related to food. This is important because the authors write about territory divergence just below. It is not clear when territories overlap and when they prove divergence.

See L35-36 when you write about low spatial overlap between species. Be consequent in your arguments.

I encourage the authors to consider the concept of territory and site in the proposed or subsequent works. Territory is an actively defended space, while site is a broader concept as it also includes feeding sites.

L20 – patterns of what?

L37 – paritioned or rather different?

L40 – partitioning or rather diverse/divided/separated? Partitioning is more suited to static phenomena, such as landscapes.

Introduction.

L53 – rather divide/separate than partition

L65-69 – this excerpt is taken out of context, I don't understand the relationship of large carnivores overlapping and relationship with humans. Please, integrate better these phenomena.

The purpose of the work and its originality should be better formulated at the end of the chapter.

Materials and methods.

L114-115 – 24 h is not only a day

Were individual bears somehow recognized? If not, how did the authors deal with the problem of double counting the same individuals while working with camera recordings?

L135 – should there be separate sentences?

I don't understand why the activities were alternately compared with parametric and nonparametric tests. Did the authors check the distribution of activities?

Results.

I suppose that this chapter starts around L167.

Figure 2 contains very valuable results that could be highlighted with colours.

Discussion, conclusions

L211 – manifestation is a very unfortunate word at this point

Discussion and conclusions sound reasonable.

L306 – research is singular

Reviewer 3 Report

This manuscript deals with the spatiotemporal behaviour amongst two similar bear species which share the same distribution range.

I feel that MS like this are interesting and provide important ecological information. Therefore, I have only some minor revisions before acceptance:

1) Line 72. I agree, but giant pandas are also endemic to China, differently from sun bears. This should be assessed.

2) Lines 81-89. I think that the part on aims should be clarified and improved. Furthermore, your predictions are lacking (in general larger species outcompete smaller ones).

3) Lines 107-116. The use of baits somehow alters the spatial behaviour and possibly also the temporal one. This part should be discussed.

4) Sampling design. Methods must be clarified. How far were camera traps from one-another?
